Anti-inflammatory and antiaging properties of chlorogenic acid on UV-induced fibroblast cell

Girsang Ermi 1 ermigirsang@unprimdn.ac.id
Ginting Chrismis N. 1
Lister I Nyoman Ehrich 1
Gunawan Kamila yashfa 2
http://orcid.org/0000-0002-5401-7794 Widowati Wahyu 3
1 Faculty of Medicine, Universitas Prima Indonesia , Medan, North Sumatera , Indonesia
2 Biomolecular and Biomedical Research Center, Aretha Medika Utama , Bandung, West Java , Indonesia
3 Faculty of Medicine, Maranatha Christian University , Bandung, West Java , Indonesia
Riaz Ahmed Kausar Begam
Electronic publication date: 2021 Jul 7
Publication date: 2021
Volume: 9
Electronic Location ID: e11419
Received 2020 Nov 18; Accepted 2021 Apr 16
Copyright: © 2021 Girsang et al.
Copyright year: 2021
Copyright holder: Girsang et al.
License: This is an open access article distributed under the terms of the Creative Commons Attribution License, which permits unrestricted use, distribution, reproduction and adaptation in any medium and for any purpose provided that it is properly attributed. For attribution, the original author(s), title, publication source (PeerJ) and either DOI or URL of the article must be cited.
License URL: https://creativecommons.org/licenses/by/4.0/

Keywords: Antiaging, Anti-inflammatory, Apoptosis, Chlorogenic acid, Ultraviolet

Funding: The Ministry of Education, Culture, Research, and Technology, Indonesia for research grant of PDUPT (Penelitian Dasar Unggulan Perguruan Tinggi) 2021 This study was supported by The Ministry of Education, Culture, Research, and Technology, Indonesia for research grant of PDUPT (Penelitian Dasar Unggulan Perguruan Tinggi) 2021. The funders had no role in study design, data collection and analysis, decision to publish, or preparation of the manuscript.

==============================
Background

Skin aging is the most common dermatological problem caused by intrinsic and extrinsic factor, such as exposure to (ultraviolet) UV rays. Chlorogenic acid (CA) is a phenolic compound which is known for its antioxidant properties against oxidative stress.

Objective

This study investigates the antiaging and anti-inflammatory properties of CA on UV-induced skin fibroblast cells.

Methods

Anti-inflammatory properties of CA were assessed by measuring inflammatory-related proteins IL-1β and TNF-α, while antiaging properties of CA were assessed by measuring reactive oxygen species (ROS), apoptosis, live and necrotic cells, and COL-3 gene expression level.

Results

Treating UV-induced skin fibroblast cells with CA decreased the level of ROS, IL-1β, TNF-α, apoptotic cells, and necrotic cells and increased live cells and COL-3 gene expression.

Conclusion

CA has the potential as the protective compound against inflammation and aging by decreasing the level ROS, pro-inflammatory cytokines IL-1β and TNF-α, apoptotic cells, and necrotic cells and by increasing live cells and COL-3 gene expression.

Introduction

Skin aging is the most common dermatological problem, especially for women. While aging could happen as a result of intrinsic mechanisms, the external causes also play a big role in causing skin aging (Baumann, 2007; Farage et al., 2008). The external causes in aging are mostly from environmental factors i.e., sun (ultraviolet), infrared and heat, smoking, air pollution, and low antioxidant diet (Vierkötter & Krutmann, 2012; Addor, 2018). Ultraviolet (UV) exposure is one of the factors that play an important role in affecting all body systems, especially for the skin as the outer barrier against the external environment. Direct exposure to UV rays could cause progressive loss of integrity and physiology manifested in visible rough, sagging, and wrinkles in the skin (Baumann, 2007). Infrared (IR) and heat induce premature skin aging with the mechanism: (1) IR stimulates the expression of Matrix Metallopeptidase-1 (MMP-1) and decreases type 1 procollagen expression in vivo. Acute IR irradiation also increases new, leaky vessel formation, and induces inflammatory cellular infiltration. (2) Heat energy increase skin temperature which increases MMP-1, -3, and -12 expression, and modulates elastin and fibrillin synthesis, resulting in the development of solar elastosis. Acute heat shock in human skin stimulates new vessel formation, recruit inflammatory cells, and causes oxidative DNA damage (Cho et al., 2009). Smoking induces human skin aging by the additive induction of MMP-1 expression (Yin, Morita & Tsuji, 2001). Air pollution can induce skin aging by generating Reactive Oxygen Species (ROS) by the particulate matter contained within the pollution (Donaldson et al., 2005). Low antioxidant diet correlates with the antioxidant that is micronutrient in various vegetable to prevent the skin aging. If people do not consume enough antioxidants, the mostly likely case is that there are wrinkles on the skin (Addor, 2018).

At the cellular level, UV exposure affects the production of free radicals known as ROS. An imbalance in ROS production causes oxidative stress, which leads to cellular or tissue changes (Davalli et al., 2016). While ROS was naturally formed by cells as a normal condition, UV exposure is known to increase the intracellular ROS level, causing oxidative stress. Oxidative stress leads to the activation of a series of signaling pathways to the apoptosis of skin fibroblast (Tanigawa et al., 2014). ROS production is also closely related with inflammation. In contrast to young individuals, aged individuals have more numbers of inflammatory cytokines, such as IL-6, IL-1β, and TNF-α (Sanada et al., 2018). The existence of these substances plays an important role in the progression of inflammatory diseases. The inhibition of oxidative stress now becomes an important approach for treating skin damage due to aging (Rinnerthaler et al., 2015; Liguori et al., 2018).

Chlorogenic acid (CA) is one of the phenolic compounds from the hydroxycinnamic family, which is known for its antioxidant properties against free radicals (Santana-Gálvez, Cisneros-Zevallos & Jacobo-Velázquez, 2017). This compound is also found in beverages prepared from herbs, fruits, and vegetables (Clifford, 1999). In vitro and in vivo analyses reported the roles of CA in mitigating oxidative and inflammatory stresses (Liang & Kitts, 2016).

CA is naturally found in snake fruit peel (Girsang et al., 2019a; Fitri, 2015). Recent in silico studies about snake fruit peel compound i.e., CA reported its antiaging and antioxidant properties (Girsang et al., 2019b). CA is also reported to have a protective effect by suppressing ROS level and to have anti-inflammatory properties against to lead (Pb) poisoning (Girsang et al., 2019c). Therefore, this study aimed to determine the antiaging and anti-inflammatory properties of CA on UV-induced fibroblast cells, being reported for its antioxidant activities.

Materials and Methods

Materials

Skin fibroblast cells (BJ cell line, ATCC no CRL-2522) were obtained from Biomolecular and Biomedical Research Center, Aretha Medika Utama, Bandung, Indonesia. Cells were grown in MEM (L0416-500; Biowest, Riverside, MO, USA) with 10% fetal bovine serum (FBS, S1810-500; Biowest, Riverside, MO, USA), 1% antibiotic/antimycotic (L0010-100; ABAM, Biowest, Riverside, MO, USA), 1% Nanomycopulitine (L-X16-100; Biowest, Riverside, MO, USA), 1% amphotericin B (L0009-050; Biowest, Riverside, MO, USA), and 0.1% gentamicin (15750060; Gibco, Waltham, MA, USA) respectively. Cells were incubated at 37 °C in a humidified atmosphere with 5% CO2. The cells with 80% confluency (n = 106 in six-well plate) were incubated with UV exposure for 75 min (37 °C, 5% CO2). The cells induced from UV exposure were treated for 4 days with CA (BP0345; Chengdu Biopurify Phytochemical) and were harvested with 0.25% trypsin-EDTA (25200072; Gibco, Waltham, MA, USA) after 4 days treatment (Widowati et al., 2016; Girsang et al., 2019c).

Measurement of IL-1β and TNF-α

The conditioned medium of the treated cells was used to measure the concentration of IL-1β and TNF-α. The measurement was conducted according to Human IL-1β ELISA Kit (Elisa Max Deluxe, 437004; BioLegend, San Diego, CA, USA) and Human TNF-α ELISA Kit (Elisa Max Deluxe, 430204; BioLegend, San Diego, CA, USA) protocol. The treatments used in this experiments were as follows: (1) normal control; (2) BJ cells + DMSO 1% (vehicle control); (3) UV-induced BJ cells (positive control); (4) UV-induced BJ cells + CA 6.25 µg/ml; and (5) UV-induced BJ cells + CA 25 µg/ml (Laksmitawati et al., 2017; Widowati et al., 2016; Widowati et al., 2019a; Lister et al., 2020; Widowati et al., 2021).

Total protein measurements

The total protein measurement was conducted to assess the IL-1β and TNF-α level by mg protein. The BSA stock was obtained by dissolving 2 mg of BSA in 1,000 μl ddH2O (A9576, Lot. SLB2412; Sigma, St. Louis, MO, USA). The BSA solution was obtained from diluting the BSA stock. The standard solution with 20 μl in number and 200 μl of Quick Start Dye Reagent 1 × (5000205; Biorad, Hercules, CA, USA) were added to each well. The plate was incubated for 5 min at room temperature. The result was read at 595 nm (Widowati et al., 2019a; Lister et al., 2020).

Measurement of ROS level

The measurement of ROS levels was assessed using DCFDA Cellular Reactive Oxygen Species Detection Assay Kit (Ab113851) according to the protocol, with slight modification by flow cytometry (MACSQuant Analyzer 10: Miltenyi, Bergisch Gladbach, Germany). The cells were harvested, counted, and added with 500 µl DCFDA working buffer. They were then stained with 25 µM DCFDA and incubated for 30 min in 37 °C, with 5% CO2. The result was analyzed using MACSQuant Analyzer 10 Flow Cytometer (Miltenyi, Bergisch Gladbach, Germany) (Widowati et al., 2014; Niocel et al., 2019; Girsang et al., 2019c; Lister et al., 2020).

Measurement of apoptotic, live, necrotic cells

The measurement of apoptotic, live, and necrotic cells was assessed by flow cytometry analysis. The cells were placed in a 2 × 6-well plate (n = 500.000) and were incubated for 2 h. The growth medium was discarded in 4 days and harvested and centrifuged for 5 min at 1,600 rpm. The pellet was added with 500 µl FACS buffer and centrifuged. The pellet was added with 100 µl FACS buffer and stained by annexin and propidium iodide (PI). The stained cells were incubated for 1 h and analyzed using flow cytometry (Lister et al., 2020; Widowati et al., 2019b).

Measurement of COL-3 gene expression

Measurement of COL-3 gene expression was performed using real-time quantitative reverse transcription polymerase chain reaction (qRT-PCR) (Thermo Fisher Scientific PikoReal 96) with SsoFast Evagreen Supermix (Bio-Rad, 172-5200). RNA isolation was performed using AurumTM Total RNA mini Kit (Bio-Rad, 732-6820) according to manufacturer’s protocol, the total RNA yield was estimated using microplate reader at 260/280 nm. RNA was used to produce cDNA using iScript cDNA synthesis (Bio-Rad, 1708890). Then the make master mix using iScript Reverse Transcription Supermix for qRT-PCR (Bio-Rad, 170-8841). The qPCR conditions were for pre-denaturation (95 °C for 5 min), denaturation (95 °C for 1 min), annealing (58 °C for 40 s), pre-elongation (72 °C for 1 min), and elongation (72 °C for 5 min). All the reaction was set for 55–90 °C melting curve and 4 °C infinite hold, in a total of 40 reaction cycles (Widowati et al., 2019b; Afifah et al., 2019).

Statistical analysis

All the data are presented as the mean ± standard deviation. The data were analyzed using Shapiro–Wilk test followed by Mann–Whitney–Wilcoxon test and independent t-test. P-values < 0.05 were considered significant.

Results

IL-1β level

The IL-1β level of the conditioned medium of the treated fibroblast cell with CA was measured using the ELISA method and compared to the negative control and UV-induced fibroblast cell (positive control). Treatment with DMSO 1% (vehicle control) was not significantly different with the negative control. The result shows a significant decreasing level of IL-1β compared to the positive control (Fig. 1A). Total protein assay was present to measure the IL-1β level by mg protein. The CA treatment showed a significant decrease in the IL-1β level compared to the positive control (Fig. 1B).

Figure 1 Effect CA towards IL-1β and TNF-α level on UV-induced fibroblast cells.

(A) IL-1β level (pg/ml), (B) IL-1β level (pg/mg protein) (I) BJ cells (negative control), (II) BJ cells + DMSO 1% (vehicle control), (III) UV-induced BJ cells (positive control), (IV) UV-induced BJ cells + CA 6.25 µg/ml, (V) UV-induced BJ cells + CA 25 µg/ml. *Data is presented as mean ± standard deviation. A single asterisk symbol (*) marks statistical difference compared to negative control group at 0.05 significance level, a single hashtag (#) marks statistical difference compared positive control at 0.05 significance level based on Tukey HSD post hoc test. (C) TNF-α level (pg/ml), (D) TNF-α level (pg/mg protein) (I) BJ cells (negative control), (II) BJ cells + DMSO 1% (vehicle control), (III) UV-induced BJ cells (positive control), (IV) UV-induced BJ cells + CA 6.25 µg/ml, (V) UV-induced BJ cells + CA 25 µg/ml. *Data is presented as mean ± standard deviation. A single asterisk symbol (*) marks statistical difference compared to negative control group at 0.05 significance level, a single hashtag (#) marks statistical difference compared positive control at 0.05 significance level based on Tukey HSD post hoc test.

TNF-α level

Treatment with DMSO 1% (vehicle control) on fibroblast cells toward TNF-α level was not significantly different compared to the negative control. Treatment with 6.25 and 25 μg/ml CA showed a significant decreasing difference compared to positive control (Fig. 1C). Total protein assay was present to measure the TNF-α level by mg protein. The CA treatment showed a significant decrease in the TNF-α level compared to the positive control (Fig. 1D). Total protein assay was present to measure the TNF-α level by mg protein. The result showed a significant decrease in the TNF-α level compared to both controls (Fig. 1D).

ROS level

The ROS level of UV-induced fibroblast cells was measured using flow cytometry with DCFDA single staining. The dot blots show the population of the analyzed cells, and the peak shows the positive ROS cells. The normal control, vehicle control, positive control, treatment with CA 6.25 µg/ml, and treatment with CA 25 µg/ml show 0.02%, 0.05%, 16,01%, 4.95%, and 6.86% positive ROS levels, respectively (Fig. 2). Both cells treated with 6.25 and 25 μg/ml showed a significant decreasing difference compared to positive control (Fig. 3).

Figure 2 The representative of dot blots of various concentration CA on UV-induced fibroblast cell toward ROS level by flow cytometry.

(A) BJ cells (negative control); 0.02%, (B) BJ cells + DMSO 1% (vehicle control): 0.05%, (C) BJ cells + UV (positive control): 16.01%, (D): UV-induced BJ cells + CA 6.25 µg/ml: 4.95%, (E): UV-induced BJ cells + CA 25 µg/ml: 6.86%.

Figure 3 Effect CA towards ROS level on UV-induced fibroblast cells.

(I) BJ cells (negative control), (II) BJ cells + DMSO 1% (vehicle control), (III) UV-induced BJ cells (positive control), (IV) UV-induced BJ cells + CA 6.25 µg/ml, (V) UV-induced BJ cells + CA 25 µg/ml. *Data is presented as mean ± standard deviation. A single asterisk symbol (*) marks statistical difference compared to negative control group at 0.05 significance level, a single hashtag (#) marks statistical difference compared positive control at 0.05 significance level based on Tukey HSD post hoc test.

Apoptosis level

The apoptosis level was analyzed using flow cytometry. The surface markers PI and annexin were used as stains. Figure 4 shows the representative of dot blots effect CA 6.25 and 25 μg/ml towards apoptosis cells. Flow cytometry analysis reveals the percentage of live cells, necrotic cells, early apoptosis, and late apoptosis of cells. Figure 5 shows the comparison between each treatment on live of cells (a), necrosis of cells (b), death of cells (early apoptosis) (c), and apoptosis (late apoptosis) (d).

Figure 4 The representative of dot blots of various concentration CA on UV-induced skin fibroblast cell toward apoptosis level by flow cytometry.

(A) BJ cells (negative control): live cells: 100%, necrotic: 0.0%, early apoptosis: 0.0%, late apoptosis: 0.0% (B) BJ cells + DMSO 1% (vehicle control): live cells: 95.14%, necrotic: 0.38%, early apoptosis: 7.86%, late apoptosis: 2.61% (C) BJ cells + UV (positive control): live cells: 38.60%, necrotic: 18.0%, early apoptosis: 16.98%, late apoptosis: 26.43% (D) UV-induced BJ cells + CA 6.25 µg/ml: live cells: 80.35%, necrotic: 0.78%, early apoptosis: 10.48%, late apoptosis: 8.39% (E) UV-induced BJ cells + CA 25 µg/ml: live cells: 85.50%, necrotic: 0.92%, early apoptosis: 6.51%, late apoptosis: 6.97%.

Figure 5 Effect CA towards apoptosis cells on UV-induced fibroblast cells.

(A) live of cells, (B) necrosis of cells, (C) death of cells (late apoptosis of cells), (D) apoptosis of cells (early apoptosis of cells) (I) BJ cells (negative control), (II) BJ cells + DMSO 1% (vehicle control), (III) BJ cells + UV (positive control), (IV) UV-induced BJ cells + CA cells + CA 6.25 µg/ml, (V) UV-induced BJ cells + CA 25 µg/ml. *Data is presented as mean ± standard deviation. A single asterisk symbol (*) marks statistical difference compared to negative control group at 0.05 significance level, a single hashtag (#) marks statistical difference compared positive control at 0.05 significance level based on Tukey HSD post hoc test.

COL-3 gene expression

The COL-3 gene expression was examined using qRT-PCR. Table 1 presents the primer and annealing temperature used for performing COL-3 gene expression, while Table 2 shows the concentration and RNA purity. The COL-3 gene expression shown in Fig. 6 revealed the CA treatment 6.25 and 25 μg/ml showed a significant increase in the COL-3 gene expression compared to the positive control.

Table 1 RT-qPCR detail of COL-3 gene.

Gene symbols	Primer Sequence (5′ to 3′)
Upper strand: sense
Lower strand: antisense	Product size (bp)	Annealing
(°C)	Cycle	References	
COL-3	5′-CCAGGAGCTAACGGTCTCAG -3′
5′-CACGGTTTCCATCTCTTCCA -3′	103	54	40	Ye et al., 2012	
β-Actin	5′-AGACCTGTACGCCAACACAG-3′
5′-TTCTGCATCCTGTCGGCAAT-3′	24	60	40	Palumbo et al., 2018	

Table 2 Concentration and purity of RNA.

		RNA concentration (ng/µL)	RNA purity
(λ 260/λ 280 nm)	
BJ cells (negative control)	190.20	2.3468	
BJ cells + DMSO 1% (vehicle control)	179.20	2.3271	
UV-induced BJ cells (positive control)	177.70	2.3179	
UV-induced BJ cells + CA 6.25 µg/ml	118.80	2.2235	
UV-induced BJ cells + CA 25 µg/ml	105.64	2.2848	

Figure 6 Effect CA towards COL-3 gene expression on UV-induced fibroblast cells.

(I) BJ cells (negative control), (II) BJ cells + DMSO 1% (vehicle control), (III) BJ cells + UV (positive control), (IV) UV-induced BJ cells + CA cells + CA 6.25 µg/ml, (V) UV-induced BJ cells + CA 25 µg/ml. *Data is presented as mean ± standard deviation. A single asterisk symbol (*) marks statistical difference compared to negative control group at 0.05 significance level, a single hashtag (#) marks statistical difference compared positive control at 0.05 significance level based on Tukey HSD post hoc test.

Discussion

UV-induced aging mechanism triggered free radicals in the form of increased intracellular ROS level, leading to oxidative stress, which then triggered inflammation leading to cell death (apoptosis). BJ fibroblast cells are the model of aging cells induced by UV (Girsang et al., 2019c). Fibroblast cells play a role in tissue granulation and scar formation during the inflammation process. The result of the anti-inflammatory test shows treating UV-induced BJ cells with CA decreased the IL-1β and TNF-α levels related to the aging model (Fig. 1). The inflammation triggered by UV rays resulted in the increase of the pro-inflammatory cytokine group IL-1β and TNF-α, which are used as inflammatory markers for anti-inflammatory assessment (Chung et al., 2009).

The presence of ROS as one of the inflammation trigger also being one of the direct UV-exposures effect in cells (Tanigawa et al., 2014). The accumulation of ROS leads to cell death known as apoptosis, which results in the decreasing level of healthy cells. The result of the assessment with fluorescence intensity as an indicator by DCFDA staining showed that CA decreased the level of ROS compared with the positive control (Figs. 2 and 3). The previous study about CA also stating the reducing ROS level induced by lead (Pb) and hydrogen peroxide (H2O2) effect by this substances (Girsang et al., 2019c; Hoelzl et al., 2010), CA has antiaging property by inhibiting enzyme related skin aging i.e., MMP-1 and human skin fibroblast elastase (HSFE) according in silico analysis (Girsang et al., 2019b).

UV exposure of BJ-cells also affects the level of apoptosis. The oxidative stress caused by UV rays promotes apoptosis in healthy cells (Kannan & Jain, 2000). Flow cytometry analysis revealed the lowest number of live cells in UV-induced BJ cells. Treatment with CA on UV-exposed cells resulted in increased number of live cells and decreased cell death, necrosis, and apoptosis (Figs. 4 and 5). The protective effect of CA toward apoptosis is also reported in rat liver with oxidative stress triggered by methotrexate (Ali et al., 2017).

While CA could potentially decrease the levels of ROS, IL-1β, and TNF-α, the measurement of COL-3 gene expression can also be conducted to determine the repairing effect of CA toward aging. COL-3 gene is related to the pattern of tendon fibril thickness and is more abundant in younger individuals than aged individuals of the animal model (Ribitsch et al., 2019). UV exposure could also decrease the COL-3 gene expression level (Chen et al., 2019). The result of this research showed the effect of UV exposure and the treatment of CA toward COL-3 gene expression level. Treatment with CA increased the COL-3 gene expression level compared to the aging model (Fig. 6). According to these results, we proposed the mechanism of anti-inflammatory and antiaging properties of CA against UV-induced BJ cells (Fig. 7).

Figure 7 Proposed mechanism of CA protective effect towards UV-induced skin fibroblast cells.

UV exposures causing an increasing level of ROS leading to an increase of pro-inflammatory cytokines IL-1β and TNF-α leading to inflammation. The presence of ROS triggered the production of caspase 9 (Casp-9) along with the production of Casp-8 by the presence of TNF-α leading to apoptosis. These two condition of inflammation and apoptosis leading to aging in cells. On another side, UV-exposures decreasing the COL-3 gene expression causing the loss of fibril pattern leading to aging as well. The presence of CA inhibits the production of ROS level thus suppressing the pro-inflammatory cytokines IL-1β and TNF-α. The presence of CA also increasing the level of COL-3 gene expression.

Conclusion

This study shows that CA has the potential as the protective compound against inflammation and aging by reducing ROS, pro-inflammatory cytokines IL-1β and TNF-α, apoptosis, and necrotic cells and by increasing live cells and COL-3 gene expression.

Supplemental Information

Supplemental Information 1 Apoptosis Level Data.

Click here for additional data file.

Supplemental Information 2 COL-3 Gene Expression Data.

Click here for additional data file.

Supplemental Information 3 IL- 1β Level Data.

Click here for additional data file.

Supplemental Information 4 ROS Level Data.

Click here for additional data file.

Supplemental Information 5 TNF-α Level Data.

Click here for additional data file.

We would also extend our gratitude to Seila Arumwardana, Hanna Sari Widya Kusuma, Cintani Dewi Wahyuni, Cahyaning Riski Wijayanti, Muhammad Aldi Maulana, and Aditya Rinaldy from Aretha Medika Utama, Biomolecular and Biomedical Research Center, Bandung for the their valuable assistance.

Additional Information and Declarations

Competing Interests

Author Contributions

DNA Deposition

Data Availability

The authors declare that they have no competing interests.

Ermi Girsang conceived and designed the experiments, analyzed the data, prepared figures and/or tables, authored or reviewed drafts of the paper, and approved the final draft.

Chrismis N. Ginting conceived and designed the experiments, prepared figures and/or tables, authored or reviewed drafts of the paper, and approved the final draft.

I. Nyoman Ehrich Lister conceived and designed the experiments, prepared figures and/or tables, authored or reviewed drafts of the paper, and approved the final draft.

Kamila Yashfa Gunawan performed the experiments, prepared figures and/or tables, authored or reviewed drafts of the paper, and approved the final draft.

Wahyu Widowati performed the experiments, analyzed the data, prepared figures and/or tables, authored or reviewed drafts of the paper, and approved the final draft.

The following information was supplied regarding the deposition of DNA sequences:

The Gen COL-3 and B-Actin sequences are available at NCBI:

COL-3, NM_000090.4; B-Actin, NM_001101.5.

The following information was supplied regarding data availability:

The raw measurements are available in the Supplemental Files.

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
