# Peer review of "Anti-inflammatory and antiaging properties of chlorogenic acid on UV-induced fibroblast cell"

_PeerJ, doi:10.7717/peerj.11419_

## Round 0.1 · original submission · Major Revisions

· Academic Editor

Major Revisions

Please address all reviewer comments.

·

Basic reporting

This study is very innovative, and hypothesis directed. I feel they did most of the experiments to prove their hypothesis. I feel the following may be necessary to be included in the paper,
1. UV could change the property of fibroblast, but how it is correlating with aging skin fibroblasts.
2. Was the supernatant collected on the last day or everyday during CA treatment?

Experimental design

I think IL-1β and TNFα results should be in one figure instead of two.
Fig 3. and Fig. 5 should be in the supplementary section.
After removing the CA treatment, how long did the fibroblast show less production of IL-1β, TNFα and ROS?

Validity of the findings

I think this study is unique and all the points they covered in this study are crucial to show the effect of CA in a aging skin.

Additional comments

This study is well designed and shows the effect of natural product on anti-inflammatory responses in skin. I have suggested few concerns. The paper seems to shed light in the area of skin immunology.

Reviewer 2 ·

Basic reporting

Chlorogenic Acid is an interesting compound to study. It appears a lot of work has already been done on it. The perspective of using UV as an insult is interesting. However multiple concerns need to be addressed. The authors must address the language in the paper and make grammatical corrections all throughout the paper.

Experimental design

Only 1 cell line has been tested which is inadequate

Validity of the findings

This study is done in 1 cell line therefore replication of findings in at least 2 more cell lines is required to increase the impact of the outcome and validity of this study.

Additional comments

Overall, it is an interesting area. However, multiple issues need to be addressed to ensure better quality of the paper. My concerns are as follows

In introduction - Mention other external factors that affect skin aging. Add more background to the pathways being addressed

Line 36 – Why especially in women? Does skin aging happen only/more in women? Or is there is a reason why it is more of a problem in women than men?

Lines 50-51, add references for this statement- The inhibition of oxidative stress now becomes an important approach for treating skin damage due to aging.

Materials – Lines 65-69 – Why are cells treated with multiple antibacterial and antimycotic agents? What were the controls and were they in a separate incubator?
How was the 75-minute UV exposure decided? What kind of UV?
It is important to make sure the study is repeated in at least 2 more cell lines for the study to be acceptable.

Line 80 – What is meant by a normal control? Also, 1% DMSO is quite high, DMSO should be in 1:1000 ratio, any reason for 1% DMSO? Also, there should be another control group of DMSO with UV.

Line 85- What is mg protein?

Bar graphs are presented in a complicated manner and the significance is difficult to understand. Also, why are 2 methods of representation needed –pg/ml and pg/mgprotein?

Line 148 and 161- State what the significant difference to controls are.

Note: DMSO affects number of apoptotic cells, reinstating the importance of reducing amount of DMSO and using that as a control

---

## Round 0.2 · accepted · Accept

· Academic Editor

Accept

All reviewer comments have been addressed.

·

Basic reporting

No comment

Experimental design

No comment

Validity of the findings

No comment

Additional comments

No comment